# A Scoping Review of Economic Evaluations to Inform the Reorientation of Preventive Health Services in Australia

**DOI:** 10.3390/ijerph20126139

**Published:** 2023-06-15

**Authors:** Rachael Taylor, Deborah Sullivan, Penny Reeves, Nicola Kerr, Amy Sawyer, Emma Schwartzkoff, Andrew Bailey, Christopher Williams, Alexis Hure

**Affiliations:** 1Health Economics and Impact, Hunter Medical Research Institute, New Lambton Heights, NSW 2305, Australia; rachael.taylor@newcastle.edu.au (R.T.); deborah.sullivan@newcastle.edu.au (D.S.); penny.reeves@hmri.org.au (P.R.); 2School of Medicine and Public Health, College of Health, Medicine and Wellbeing, University of Newcastle, University Drive, Callaghan, NSW 2308, Australia; 3Health Promotion, Mid North Coast Local Health District, Coffs Harbour, NSW 2450, Australia; nicola.kerr@health.nsw.gov.au (N.K.); amy.sawyer@health.nsw.gov.au (A.S.); emma.schwartzkoff@health.nsw.gov.au (E.S.); 4Research and Knowledge Translation Directorate, Mid North Coast Local Health District, Port Macquarie, NSW 2444, Australia; andrew.bailey@health.nsw.gov.au (A.B.); christopher.williams1@health.nsw.gov.au (C.W.); 5University Centre for Rural Health, School of Health Sciences, University of Sydney, 61 Uralba Street, Lismore, NSW 2480, Australia

**Keywords:** economic evaluation, health services, prevention, local public health services, review, value-based healthcare

## Abstract

The Australian National Preventive Health Strategy 2021–2030 recommended the establishment of evidence-based frameworks to enable local public health services to identify strategies and interventions that deliver value for money. This study aimed to review the cost-effectiveness of preventive health strategies to inform the reorientation of local public health services towards preventive health interventions that are financially sustainable. Four electronic databases were searched for reviews published between 2005 and February 2022. Reviews that met the following criteria were included: population: human studies, any age or sex; concept 1: primary and/or secondary prevention interventions; concept 2: full economic evaluation; context: local public health services as the provider of concept 1. The search identified 472 articles; 26 were included. Focus health areas included mental health (*n* = 3 reviews), obesity (*n* = 1), type 2 diabetes (*n* = 3), dental caries (*n* = 2), public health (*n* = 4), chronic disease (*n* = 5), sexual health (*n* = 1), immunisation (*n* = 1), smoking cessation (*n* = 3), reducing alcohol (*n* = 1), and fractures (*n* = 2). Interventions that targeted obesity, type 2 diabetes, smoking cessation, and fractures were deemed cost-effective, however, more studies are needed, especially those that consider equity in priority populations.

## 1. Introduction

Driven by the unsustainable burden of chronic disease, a shift is occurring within healthcare systems globally from curative, treatment-focused health towards preventive health. The preventive health approach aims to improve the health and well-being of a population by “reducing the likelihood of a disease or disorder, interrupt or slow the progression or reduce disability” [1]. In conjunction with this shift is an emphasis on health system changes that align with value-based healthcare. While there is no universal definition of what constitutes “value”, fundamentally the approach attempts to deliver financially sustainable healthcare, as opposed to cost reduction, while keeping the needs, experiences, and outcomes that matter to the patient at the core [2].

Integration and adoption of preventive health into existing health systems require leadership and support for significant health service reorientation. Indeed, the World Health Organization (WHO) recognises that of the five actions identified in the Ottawa Charter, reorientation of health services has been the most challenging [3]. Several recommendations have been made on how nations can develop strategies that positively influence the reorientation of health services, emphasising that development and design be contextual; achievable within the current health system, resource, and economic capabilities; and aligned with local values and preferences [3].

Like many other high-income countries, Australia’s current health systems focus heavily on treatment of illness and disease, with issues of access to healthcare and health inequity. The National Preventive Health Strategy 2021–2030 aims to rebalance the health system through a long term, systems-based approach [4]. This strategy acknowledges that the burden of ill health is not shared equally among the Australian community, and any service reorientation planning must include concerted efforts to reduce disparities and improve health outcomes among priority populations. In Australia, these groups include, but are not limited to, Aboriginal and Torres Strait Islander people, culturally and linguistically diverse (CALD), lesbian, gay, transgender, queer or questioning, intersex, and/or sexuality and gender diverse people (LGBTQI+), people with mental illness, people of low socioeconomic status, people with disability, and rural, regional, and remote communities.

One of the policy goals identified within the National Preventive Health Strategy is the establishment of local prevention frameworks [4]. Ideally, these frameworks are evidence based, incorporating the elements of value. Economic evaluations can assist local public health services identify strategies and interventions within their local framework that demonstrate cost-effectiveness, representing value for money. Such evaluations need to consider local contextual factors, such as resource allocation, and what is within jurisdictional purchasing power [5].

Both the Australian Institute of Health and Welfare and the Productivity Commission have highlighted the need to ensure sustainability of Australia’s health expenditure by addressing the growing disparity in investment in preventive health compared to clinical services, specifically noting that despite the potential for significant returns from investments into preventive health, the field suffers from a relative lack of funding [6,7]. This review specifically seeks to identify where there is evidence of cost-effectiveness or returns on investment in preventive health.

A scoping review is a type of evidence synthesis that can be used to systematically map the scope, characteristics, and findings in an area, which is useful for identifying priority areas for future research, policy, and practice. Therefore, this type of research design is highly appropriate for summarising the evidence base to support the development of local prevention frameworks. To our knowledge, no scoping reviews have been conducted that have identified and mapped the evidence for preventive health strategies for multiple health risk factors and/or health conditions for predominately high-income countries. This review is important to provide a synthesis of relevant findings and draw conclusions based on the strength of the evidence to support translation. The aim of this scoping review was to identify and synthesise the available evidence from systematic reviews on the cost-effectiveness of preventive health strategies with relevance to local public health services, to inform the reorientation of preventive health services and delivery of value-based healthcare.

## 2. Materials and Methods

This scoping review was conducted in accordance with JBI methodology for scoping reviews [8] and reported using the PRISMA-ScR Reporting Standards (Table A1) [9] (Appendix A, Table 1). The protocol for the scoping review is provided in Appendix A, Table A2. Due to the exploratory nature of scoping reviews and the breadth of preventive health, a review of reviews approach was used [10], searching for publications that include high-level aggregate data and/or an evidence synthesis of primary trials. The purpose was to extract evidence that has already been synthesised and identify cost-effective focus areas for intervention in preventive health.

### 2.1. Definition of Key Terms

#### 2.1.1. Types of Preventive Health

Types of preventive health were based on the National Preventive Health Strategy definitions, which represents a continuum spanning from wellness to ill health [4]. Primordial prevention, as defined by the strategy, is focused on the wider determinants of health by addressing the social and environmental factors across the entire population through strategies such as taxation, regulation, and infrastructure [4]. Primordial strategies require multilevel, multisectoral collaboration and investment and therefore fall outside the remit of local public health services. Primary prevention is focused on reducing risk factors to prevent ill health before it occurs through population-level strategies such as vaccination and targeted strategies for high-risk individuals, such as people with high blood pressure, low physical activity, poor dietary intake, or overweight/obesity [4]. Secondary prevention is focused on identifying individuals at high risk of ill health as well as early detection and management of a disease or disorder to either prevent or slow the long-term effects, using strategies such as health screening and counselling and education programmes [4]. Both primary and secondary health promotion are within the remit and purchasing power and embedded in service-level agreements of local public health services. Tertiary prevention focuses on managing established disease or disorder to maximise functional ability [4]. Quaternary prevention focuses on reducing harm from medical interventions used to manage a disease or disorder [4].

Health promotion is the process of “empowering people to increase control over their health and its determinants through health literacy efforts and multisectoral action to increase healthy behaviors” [11]. Disease prevention and health promotion share considerable overlap in goals and functions. The WHO characterise disease prevention services as those primarily concentrated within the healthcare sector, whereas health promotion services depend on intersectoral actions and/or are concerned with the social determinants of health.

#### 2.1.2. Economic Evaluation and Evaluation Methods

For this scoping review, economic evaluation was defined as the “comparative analysis of alternative courses of action in terms of both their costs and consequences” [12]. There are several economic evaluation methods that can be used to evaluate cost-effectiveness. While measurement of cost is common to all methods, measurement and valuation of outcomes vary.

Cost-effectiveness analysis (CEA) measures outcomes in natural health units such as deaths prevented, units of blood pressure, or minutes of physical activity. Cost–utility analysis (CUA) measures outcomes in quality-adjusted life years (QALYs), disability-adjusted life years (DALYs), or health-adjusted life years (HALYs), combining survival with quality of life, measured using preference-based, multiattribute utility instruments [13]. Both CEA and CUA compare alternatives using a summary measure, incremental cost-effectiveness ratio (ICER). ICERs can be compared against a pre-determined cost-effectiveness threshold, recognised as a willingness to pay for a QALY [14]. While thresholds vary between countries and debate surrounds their origins and limitations [15], ICERs provide decision makers with a benchmark to guide value-based decisions and some level of comparability when allocating scarce resources. The United Kingdom has published their willingness to pay threshold for a QALY as from GBP 20,000–30,000 [16]. However, empirical evidence suggests in practice the true threshold sits at GBP 13,000 [17]. In the United States, ICER thresholds range from USD 50,000 to USD 200,000 [18]. The WHO have a published threshold, generally for low- and middle-income countries, set at one to three times the per capita gross domestic product [19]. While Australia has no explicitly stated or public threshold, empirical studies have reported thresholds around AUD 28,000 based on decision-making patterns for pharmaceutical reimbursement [20].

Cost–benefit analysis (CBA) values outcomes in monetary terms with an action deemed cost-effective if the benefit to cost ratio is greater than 1. Cost–consequence analysis (CCA), a form of CBA, includes monetised outcomes where available alongside non-monetised outcomes reported in natural units, allowing decision makers to assess value, albeit subjectively. Cost-minimisation analysis (CMA) is a method commonly associated with non-inferiority trials. Where outcomes are shown to be statistically equivalent between comparators, the analysis is constrained to looking at differences in cost only and the alternative with the lowest cost is favourable. Return on investment (ROI), while not strictly a comparative analytical approach, is the monetary benefit minus cost expressed as a proportion of the cost [21]. For example, a programme that spends AUD 1 and saves AUD 9 in future spending has an ROI of 800%. Social return on investment (SROI) and social cost–benefit analysis (SCBA) are emerging approaches, which attempt to monetise outcomes not typically captured, such as wider social and environmental outcomes [22].

### 2.2. Search Strategy

The search strategy was developed and tested in consultation with a research librarian (JB) following the mixed method Population, Concept, Context (PCC) framework [23]. The search strategy included grey literature to find reviews of economic evaluations, relevant to local public health services, contained within reports and government documents, and not typically located in peer-reviewed publications.

An initial search strategy was piloted in MEDLINE with iterative screening of the first 100 titles and abstracts until the search terms were set (Table 1). The final search was performed in the following databases: MEDLINE, Embase, APO, and MedNar, for review articles published between 2005 and February 2022. The search of the academic literature was limited to 2 databases for pragmatic reasons. The year 2005 was chosen as data from PubMed indicated that 79% of articles related to preventive health interventions and economic evaluation were published after this date (Table A3). Furthermore, health economic evaluations were not vigorously reported until the introduction of the International Society for Pharmacoeconomics and Outcomes Research Task Force guidance for economic evaluation alongside clinical trials which occurred in 2005 [24]. Australian health economic and tertiary institution websites were manually searched for the same period, using search filters/terms defined by the institutions’ own search engines. The full search strategy is available in Appendix A (Table A4 and Table A5). Citations identified by the search were collated and uploaded into EndNote X9 [25] and duplicates removed.

### 2.3. Selection of Articles

Articles were included if they described a review of economic evaluations for primary- and/or secondary-level prevention interventions within or relevant to a local public health service setting. Health promotion was included only when the intervention fell within the resourcing of local public health services. Economic evaluations were restricted to full evaluations, excluding partial economic evaluations (e.g., micro-costings), methodological reviews, or economic frameworks. Reviews were excluded if the authors identified the preventive health strategies as primordial, tertiary, or quaternary. Even though local public health services engage in tertiary preventive health strategies, the scope of this review was focused on primary and secondary prevention, with a view to reorienting health services from illness to wellness.

Clinical treatments, such as medical devices and pharmacotherapy for established disease, were excluded, except for therapies specifically for reducing tobacco use and nicotine addiction. In the absence of a stated level of prevention classification, the National Preventive Health Strategy was used as a reference point [4].

To increase generalisability to the Australian context, reviews of studies predominantly conducted in high-income countries, as defined by the World Bank for 2022 fiscal year [26], were included. Where reviews included studies in both high- and middle-income countries, a cut-off of ≤25% of all studies being from middle-income countries was applied. Reviews of low-income countries, global data, or aggregates of large regions (such as the European Union) were excluded. Non-English publications were also excluded due to resource constraints. Table 2 outlines the full inclusion and exclusion criteria applied for the screening of articles.

### 2.4. Evidence Screening and Selection

Pilot screening was conducted on titles and abstracts by two independent reviewers (DS, AH) for assessment against the initial eligibility criteria, with discrepancies resolved and revisions made to clarify eligibility criteria (Table 2). The remainder of the screening and selection process was undertaken primarily by one reviewer (DS), with 20% screened in duplicate by a second reviewer (AH); agreement was high at >95%.

### 2.5. Data Extraction and Synthesis

Data extraction was completed by two reviewers (DS, RT) with 20% screened by a third reviewer (AH). Data extraction was conducted within Microsoft Excel software (v16). Characteristics of the studies in each review were extracted, including the number of countries represented, date range of publication, review aim, population included, median sample size, and type of prevention intervention. Extracted data were then descriptively or quantitatively summarised (i.e., median, minimum, maximum). Detailed mapping of the priority populations, as defined by the National Preventive Health Strategy [4], included in each of the reviews was undertaken.

The economic evaluation characteristics and results of each review were extracted, including the number of economic evaluations, method of analysis, study design, valuation of outcomes, and key economic findings. There is debate in the literature regarding the value of meta-analysis for economic evaluations that are heterogeneous [27,28]; therefore, a narrative approach was taken to summarise the study findings. Intuitive conclusions were drawn from the economic evidence within each focus area, classified into the following categories: cost-effective, not cost-effective, lack of evidence, and unclear, based on the criteria reported in Table 3. Where there were multiple reviews concluding cost-effectiveness within the same focus area (e.g., type 2 diabetes), the individual studies were compared across reviews to identify overlap and avoid misrepresenting the strength of evidence.

### 2.6. Risk of Methodological Bias Appraisal of the Body of Evidence

In accordance with scoping review methods, appraisal of the risk of economic methodological bias in the included reviews was not conducted [9]. However, methodological appraisals conducted within the reviews, including assessment tools used, were extracted as part of the study characteristics.

## 3. Results

A total of 472 records were identified during the initial search with 192 duplications, returning 317 unique articles for screening. At title and abstract screening, 198 records were excluded, and 1 article not able to be retrieved. One hundred and eighteen full text articles were assessed for eligibility. A total of 26 systematic reviews were included in full data extraction. The results of each stage are illustrated in Figure 1.

### 3.1. Characteristics of Included Reviews

The characteristics of the 26 included systematic reviews are described in Table 4. The systematic reviews were predominately (19 of 26) published between 2015 and 2021. Across the systematic reviews, there were 674 economic evaluation studies, conducted in high-income (*n* = 22 countries) and middle-income (*n* = 10) countries (Figure A1). All systematic reviews (26 of 26) had ≥90% of included studies from high-income countries. Many of the reviews (*n* = 18) included studies conducted in Australia. Vos et al. [29] exclusively included 150 preventive health interventions that were modelled with the Australian population in 2003 as well as 21 interventions for the Australian indigenous population. The authors of these reviews did not report there was a significant difference in the findings of Australian studies versus other high-income countries. The sample population of the studies in the systematic reviews included universal (*n* = 7), adults (*n* = 13), adults and adolescents (*n* = 3), and children (*n* = 3). The systematic reviews that included priority populations are summarised in Table 5. The highest proportion of systematic reviews in priority populations included people with disabilities (11 of 26 reviews) and mental illness (10 of 26). Very few systematic reviews included Indigenous people (1 of 26) and LGBTQI+ (1 of 26). Only half (13 of 26) reported the sample sizes of the included studies; for these the median sample was 911 individuals (minimum = 196, maximum = 1,216,000).

Most systematic reviews (22 of 26) aimed to identify studies related to specific interventions (e.g., psychological) for a risk factor or condition (e.g., psychotic experiences). The remaining four aimed to identify economic evaluations in public health, without limiting to any particular focus area. Within reviews, studies were often grouped by characteristics such as population, intervention sub-types, intervention approach (i.e., universal vs. targeted), method of economic evaluation, methodological quality, and type of economic outcomes reported. More than half of the reviews (16 of 26) included interventions that targeted primary prevention. Prevention focus areas included mental health (*n* = 3 reviews), obesity (*n* = 1), type 2 diabetes (*n* = 3), dental caries (*n* = 2), public health (*n* = 4), chronic disease (*n* = 5), sexual health (*n* = 1), immunisation (*n* = 1), smoking cessation (*n* = 3), reducing alcohol (*n* = 1), and fractures (*n* = 2).

### 3.2. Economic Evaluation Methods

The economic evaluation methods and key findings are described in Table 6. The median number of economic evaluation studies included in the systematic reviews was 16 (minimum = 1, maximum = 150). Economic analysis methods included CEA, CUA, CBA, CCA, SROI, and ROI.

### 3.3. Risk of Methodological Bias of the Evidence from the Systematic Reviews

Twenty of twenty-six systematic reviews used an assessment tool to appraise the risk of economic methodological bias (Table 6). The quality assessment tools used included: the Drummond Critical Appraisal of Economic Evaluations Checklist (*n* = 6) [52], guidelines for authors and peer reviewers of economic submissions to the British Medical Journal (*n* = 4) [52], Krlev et al.’s framework (*n* = 2) [53], Consolidated Health Economic Evaluation Reporting Standards (CHEERS) Checklist (*n* = 2) [54], Assessing Cost Effectiveness (ACE) Study Priority-Setting Checklist (*n* = 1) [29], Community Guide protocol for economic evaluations (*n* = 1) [55], Consensus on Health Economics Criteria list (*n* = 1) [56], Quality Assessment Tool for Quantitative Studies (*n* = 1) [57], Quality of Health Economic Studies Instrument (*n* = 1) [58], National Institute for Health and Care Excellence (NICE) quality appraisal checklist for economic evaluations (*n* = 1) [59], and Philips’s Checklist (*n* = 1) [60]. Eleven systematic reviews [18,21,22,30,33,35,38,40,42,43,48] concluded that at least 70% of studies were highly rated for their methodological quality (Table 7). Limitations of the evidence commonly related to the use of a short time horizon, limited perspective for the economic analysis, and a higher proportion of studies from the United States.

### 3.4. Cost-Effective or Not Cost-Effective?

The categorisation of the cost-effectiveness of the interventions by health area for the included systematic reviews is reported in Table 7 and summarised below. Details about the type of interventions included in the reviews are provided in Appendix A Table A6.

#### 3.4.1. Mental Health

Three systematic reviews [18,30,31] evaluated the economic evidence of mental health interventions. Le at al. [30] reviewed primary intervention studies (*n* = 65) for mental health disorders and mental health promotion across all life stages. The main types of interventions included were cognitive behavioural therapy, standard psychological intervention, school-based interventions, parenting interventions, and screening plus psychological interventions. Li et al. [34] classified 64% of the interventions as “unclear” since the health benefits associated with the intervention were at a higher cost. Thirty four percent of the interventions were classified as “favoured” which focused on children, adolescents, or adults and targeted the prevention of depression and suicide or promotion of mental health. The cost-effectiveness of these interventions was classified as not clear due to the broad scope of the systematic review which considered interventions that targeted multiple mental health conditions across different life stages.

Park et al. [31] reviewed secondary intervention studies (*n* = 11) for physical health promotion in adults and older adults with mental health disorders. There was a wide range of interventions that were included, such as cognitive behavioural therapy, physical exercise, and smoking cessation programmes. The cost-effectiveness of these interventions was classified as not clear. While there were 11 studies in the review the studies were too heterogeneous to draw conclusions on cost-effectiveness.

Soneson et al. [18] reviewed secondary prevention interventions (*n* = 2) for psychotic experiences in adolescents and adults. There was insufficient evidence to determine the cost-effectiveness of cognitive behavioural therapy interventions; the two articles identified were based on data from a single RCT.

#### 3.4.2. Obesity

One systematic review [32] evaluated the long-term (≥40 years) impact of primary prevention intervention studies (*n* = 16) for obesity for all life stages. The main types of interventions included diet, physical activity, and lifestyle. Lehnert et al. [32] reported that 81% of behavioural and 75% of community interventions were cost-effective or cost-saving. In particular, this systematic review found that seven of nine lifestyle interventions were cost-effective [32]. These interventions were predominately (83%) in adults and the economic evidence for interventions that targeted children was not favourable. Interventions targeting adults were therefore classified as cost-effective, while interventions in children were classified as lacking evidence as only three studies were included. Nine of sixteen studies included in the review were based on economic evidence from the Australian ACE study on prevention of obesity, which overlapped with studies included in the systematic review by Vos et al. [29] included in this scoping review; however, this did not change the interpretation of obesity prevention being cost-effective.

#### 3.4.3. Type 2 Diabetes

Three systematic reviews [33,34,35] evaluated the economic evidence of type 2 diabetes interventions for adults. Interventions were classified as cost-effective across all three systematic reviews. Glechner et al. [33] reported that 94% of studies (*n* = 14) found that diet and physical activity intervention studies were cost-effective. Li et al. [34] reported that 85% of diet and physical activity intervention studies (*n* = 22) were cost-effective. Group-based programmes were found to be more cost-effective compared with individual-based programmes [33]. Zhou et al. [35] reported that lifestyle interventions targeting diet and physical activity were the most cost-effective interventions, followed by metformin interventions. The median ICERs for group-based interventions were less than half of those for individual-based interventions [35]. In total there were 64 studies included across the 3 systematic reviews; 18 studies (28%) overlapped between reviews. There were sixteen studies that overlapped between two reviews, and two studies between three systematic reviews; this did not change the interpretation of type 2 diabetes prevention being cost-effective.

#### 3.4.4. Dental Caries

Two systematic reviews [36,37] evaluated the economic evidence of dental caries interventions in children. Anopa [36] reviewed primary prevention intervention studies (*n* = 16) for dental caries in pre-school-aged children. The main types of interventions included were multicomponent interventions, fluoride treatment, molar sealant, and oral hygiene and diet education. The cost-effectiveness of these interventions was classified as unclear since only 40% and 50% of studies that conducted CEA and CBA, respectively, reported the interventions to be cost-effective. Fraihat et al. [37] also reviewed prevention studies (*n* = 19) but for both pre-school-aged and primary-aged children. A wide variety of interventions were included and sub-group analyses indicated that primary prevention interventions were only effective in reducing incremental cost for children older than six years (*n* = 4) and were not cost-effective for children less than six years old (*n* = 14). These interventions were classified as not clear due to the mixed findings.

#### 3.4.5. Public Health

Four systematic reviews [21,22,38,39] evaluated the economic evidence of public health interventions. A wide variety of interventions were included such as physical activity, substance misuse, child behavioural management, community-based programmes, and healthy lifestyle interventions. These reviews were broad in scope and included interventions that targeted multiple health conditions across different life stages. As the studies included were too heterogeneous to draw conclusions, cost-effectiveness was classified as unclear for the four systematic reviews.

#### 3.4.6. Chronic Disease

Five systematic reviews [29,40,41,42,43] evaluated the economic evidence for chronic disease prevention. Three systematic reviews [29,40,41], including one review in which the interventions were modelled exclusively on the Australian population, were broad in scope and included interventions that targeted multiple health conditions across different life stages. Therefore, the cost-effectiveness of interventions was assessed as unclear for these three systematic reviews. Mattli et al. [42] reviewed physical activity intervention studies (*n* = 12) for chronic disease in adults. These interventions were classified as not cost-effective since 82% of the studies reported an ICER above the cut-off defined by Mattli et al. [42]. The systematic review of lifestyle interventions for chronic disease prevention in adults by Pennington et al. [43] only included three studies; therefore, it was classified as lacking evidence.

#### 3.4.7. Sexual Health

One systematic review [44] evaluated primary and secondary intervention studies (*n* = 31) for sexually transmitted infections and human immunodeficiency virus. The majority (25 of 31 studies) of the included studies assessed the cost-effectiveness of different screening approaches for chlamydia trachomatis. The cost-effectiveness of these interventions was classified as unclear, because findings were mixed with 52% of the studies indicating that chlamydia trachomatis screening is cost-effective for adults less than 30 years of age.

#### 3.4.8. Immunisation

One systematic review [45] evaluated economic evidence of influenza vaccination studies (*n* = 8) for children. Influenza vaccines were classified as cost-effective since all included studies in the systematic review concluded that vaccinations, specifically the quadrivalent formulation, were cost-effective. Six of eight studies were funded by pharmaceutical companies or employees were co-authors of articles.

#### 3.4.9. Smoking Cessation

Three systematic reviews [46,47,48] evaluated the economic evidence of smoking cessation interventions for adults. Cheung et al. [46] reviewed online smoking cessation interventions in the Netherlands. There was a lack of evidence to draw conclusions about the cost-effectiveness of these interventions, as only two eligible studies were identified.

The two other reviews on smoking cessation were classified as cost-effective. Lee et al. [47] reviewed adult inpatient smoking cessation interventions (*n* = 9) and found they were highly cost-effective and the degree of cost-effectiveness might not be related to the components of the programme or methodological variations in the cost-effectiveness analysis. Mahmoudi et al. [48] reviewed non-nicotine therapies for smoking cessation (*n* = 10) and found varcenicline (a drug that blocks nicotine from triggering the release of dopamine) was clinically superior and cost-saving compared to bupropion (a drug used to balance dopamine levels when nicotine is excreted from the body) in most cost-effectiveness models. Variations in time horizon, cost of bupropion, efficacy of either drug, age, and the incidence of smoking-related disease were noted as factors that could change the interpretation of results.

#### 3.4.10. Reducing Alcohol

One systematic review [49] evaluated the economic evidence of telehealth medicine for alcohol abuse, addiction, and rehabilitation. There was a lack of evidence to draw conclusions about the cost-effectiveness as only one study was included in the review.

#### 3.4.11. Fractures

Two systematic reviews [50,51] evaluated the economic evidence for a fracture liaison service programme and it was categorised as cost-effective. Ganda et al. [50] reported that four of four studies on identification, assessment, and treatment of patients as part of the service showed it was cost-saving or cost-effective. One study on identification, assessment, and then referral for treatment by a primary care physician also showed it was cost-effective. Wu et al. [51] reported that the fracture liaison service was cost-effective regardless of the intensity of the service delivery or the country of the implemented service. In total there were twenty-four studies included across the two systematic reviews; one study (4%) was identified in both systematic reviews; however, this did not change the interpretation of fracture prevention being cost-effective.

## 4. Discussion

This review used a systematic approach to map the best of the available evidence regarding the cost-effectiveness of preventive health strategies. The accessibility of economic evidence nationally and internationally and the health economics knowledge and skills of health decision makers are significant barriers for the use of economic evidence in decision making [61,62]. Therefore, the scope, characteristics, and findings of research in this area were synthesised and summarised to provide visibility to existing evidence for local public health services. This can be used in priority setting and to inform the development of local prevention frameworks that support the reorientation and delivery of value-based healthcare.

Our evidence synthesis of 26 systematic reviews found obesity (in adults), type 2 diabetes, smoking cessation, immunisation, and fracture prevention were cost-effective preventive health areas, based on existing evidence. For more than half (65%) of the reviews there was either not enough evidence to draw conclusions or the findings were unclear. This review provides clear guidance for where further economic evaluations are needed within preventive health.

In Australia, the National Preventive Health Strategy 2021–2030 identified seven focus areas for the prevention of chronic disease which include nutrition, physical activity, tobacco, immunisation, cancer screening, alcohol and other drug use, and mental health [4]. These focus areas were given priority to boost prevention in the first years of the strategy as cancer, mental health, and substance abuse disorders were the leading national burden of disease groups in 2015 [63]. Tobacco use, overweight and obesity, and dietary risks are the main modifiable factors contributing to the national disease burden [63]. Cadilhac et al. [64] reported that by targeting five risk factors (poor diet, physical activity, tobacco use, excessive alcohol consumption, and overweight and obesity), cost savings of AUD 2334 million over the lifetime of the Australian adult population could be achieved. While the strategy aims to promote health benefits particularly in communities with health inequalities and generate health gains across all life stages through impactful and coordinated initiatives within these focus areas [4], local public health services are required to implement state-level frameworks that are not well aligned with the strategy.

The economic methodology of the studies included in the systematic review varied widely based on modelling approach (e.g., trial-based analysis, modelled dichotomy economic evaluations), time frame, perspective of analysis, and study context. This heterogeneity was acknowledged within various systematic reviews [35,40,47] and precluded meta-analysis, therefore a narrative approach was taken. There are pros and cons for each of the modelling approaches. An advantage of “trial-based analysis” is that the relative treatment effect is based on a study design that minimises the risk of selection bias through use of randomisation. However, it is argued that “trial-based analysis” represents only a partial form of analysis because the study design only compares a limited number of interventions, the length of follow-up is shorter than what is required for economic analysis, it may not be relevant to the decision context, it does not incorporate all evidence that is available, and the decision uncertainty can only be quantified based on evidence from the trial (a single input) [65]. “Modelled” dichotomy economic evaluations have the advantage of being able to fully characterise decision uncertainty by combining data from multiple inputs including clinical efficiency data from trials. Two systematic reviews [22,38] evaluated studies that used SROI and SCBA which are recently adopted approaches for conducting economic evaluations. These systematic reviews identified that SROI and SCBA studies have predominately been implemented in the United Kingdom and published in the grey literature [22,38]. The methodological weaknesses (e.g., use of estimated or subjective parameters, assumptions are required) associated with these approaches have been acknowledged as a contributing factor for the lack of published studies in the peer-reviewed literature.

This scoping review has several strengths. Firstly, health service stakeholders co-created the design, conduct, analysis, interpretation, and drafting of the manuscript. The scoping review was conducted and reported in accordance with the PRISMA-ScR Reporting Standards. The review synthesised the highest level of evidence (systematic review) and included preventive health strategies that targeted any type of health problem across different life stages. The review also has some limitations that should be considered. The search of the academic literature was limited to two databases: MEDLINE and Embase. The search terms were not exhaustive and included studies were limited to those published in English between 2005 and 2022. Reviews were only included if primary and secondary preventive health strategies were relevant to local public health services and studies were predominately conducted in high-income countries. Studies were conducted in a wide variety of healthcare settings which may limit the generalisability of the findings to other local public health services. However, reviews in the type 2 diabetes focus area included the prescription of metformin as the study intervention which is a treatment rather prevention and is outside the scope of local public health services. These reviews were included to avoid excluding diabetes prevention programme interventions as the study intervention or comparator which is relevant to local public health services.

Preventive health interventions, such as sustained behaviour change compared with clinical interventions, require a long-term follow-up period or modelled dichotomy economic evaluations to observe the anticipated health gains. Many of the trial-based analyses were reliant on interventions with a short-term follow-up period and, therefore, the economic benefits were limited to intermediate indicators. The perspective of the analysis varied considerably by the studies included in the systematic reviews. This may reflect the lack of consensus on the recommendations from the study perspective provided by national healthcare economic evaluation guidelines [65,66]. Weise et al. [67] recently reviewed the assessment approaches for transferability and recommended that the assessment methods chosen should be relevant to the health area and the context of the decision making.

### 4.1. Implications

In this review, the following preventive interventions were concluded to be cost effective: adult obesity (behavioural and community interventions), type 2 diabetes (lifestyle interventions), smoking cessation (adult inpatient programme and non-nicotine therapies), immunisation, and fracture prevention (fracture liaison service programme). However, to enable the use of economic evidence to inform public policy agendas and political prioritisation, local public health services may still want to consider if systematic review evidence is transferable to their local context prior to setting policies and implementing the evidence. One study by Nystrand et al. [64] examined delivery differences, feasibility of implementation, costings, and intervention outcomes when assessing the potential transferability of systematic review evidence of the cost-effectiveness of public health interventions targeting the use of alcohol, illicit drugs, and tobacco, as well as problematic gambling behaviour. While this approach may have utility, it is resource intensive. In comparison, Welte et al. [68] and Goeree et al. [69] developed user-friendly decision charts and a classification system that indicates transferability factors and approaches for improving transferability to support decision-making processes.

### 4.2. Future Research Directions

For many (65%) of the systematic reviews, the authors of the current scoping review concluded there was not enough evidence or the evidence was unclear regarding the cost-effectiveness of the interventions. This highlights the need for further research so more definitive conclusions can be drawn regarding the economic evidence for preventive health interventions. Greater consideration is also needed for priority populations in future research, especially for Indigenous people and LGBTQI+. Wider determinants of health such as social, environmental, structural, economic, cultural, biomedical, commercial, and digital factors prevent these priority populations from having fair and just opportunities to attain the highest level of health and lead to inequity [4]. The United National 2030 Agenda for Sustainable Development strives to “leave no one behind”; this commitment is reflected in 17 Sustainable Development Goals (SDGs) [70]. A call to achieve health equity is implied in SDG3 “ensure healthy lives and promote well-being for all at all ages” [70]. The National Health Strategy also aims to address health equity in priority populations [4]. Therefore, ensuring that equity is considered in future research is important as this is a high priority for local public health services for informing policy. However, cost-effectiveness analysis was primarily designed to optimise efficiency in the allocation of healthcare resources without considering health equity [71]. This prevents local public health services from understanding if there are any trade-offs between efficiency and equity. Alternative methods to the traditional cost-effectiveness analysis have been developed, such as equity-informative cost-effectiveness analysis and distributional cost-effectiveness analysis, which is an important step for the consideration of health equity in future economic evaluations [72,73,74,75].

Conducting prospective economic evaluations in which costs are recorded for the intervention design and local adaptation, implementation, and scale-up will be essential. Sohn et al. [76] has provided a conceptual framework consisting of three phases: design, initiation, and maintenance, to assist researchers in assessing implementation costs. Jalai et al. [77] recently reviewed statistical approaches for addressing missing data when conducting prospective economic evaluations alongside clinical trials. This evidence will assist local public health services in understanding the application of potential interventions for use in different contexts.

## 5. Conclusions

This scoping review identified a large amount of evidence from systematic reviews on the cost-effectiveness of preventive health strategies, however, for most reviews there was a lack of evidence or the evidence was unclear. Interventions targeting obesity, type 2 diabetes, smoking cessation, and fractures were found to be cost-effective. We found limited evidence related to equity in priority populations. Local contextual factors need consideration in the translation of these findings into practice, including local public health services.

## Figures and Tables

**Figure 1 ijerph-20-06139-f001:**
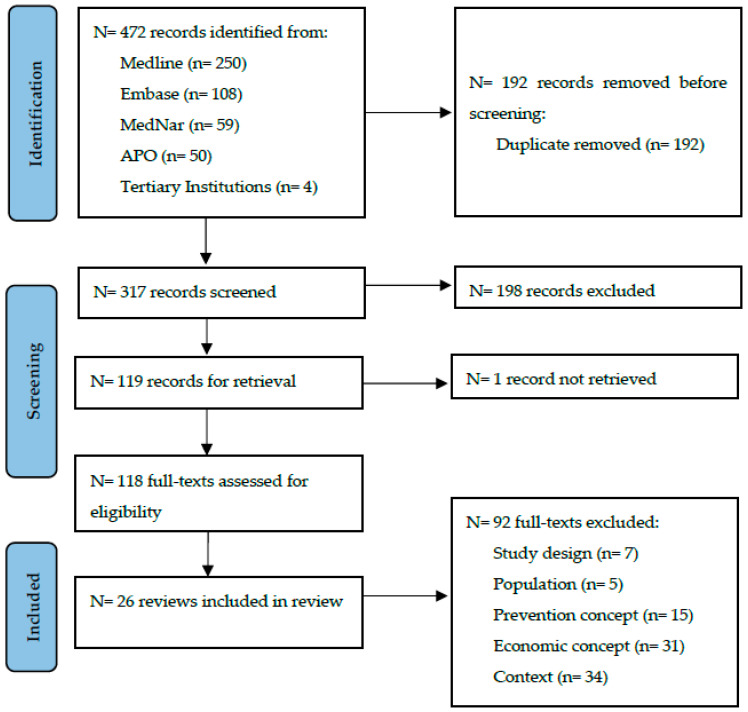
Preferred Reporting Items for Systematic Reviews and Meta-Analyses (PRISMA) depicting the identification, screening, and inclusion of reviews.

**Table 1 ijerph-20-06139-t001:** Keywords included in search strategy.

Construct	Search Terms
Study design	Review
Prevention	Primary prevention, secondary prevention, health promotion
Economic	Cost effectiveness, value for money, cost benefit analysis, cost utility analysis, cost consequence analysis, return on investment, social return on investment, cost minimisation analysis, economic evaluation, cost saving, cost efficient
Context	Healthcare service, public health ^a^

^a^ “Local public health services” was not used as search term but introduced during screening process.

**Table 2 ijerph-20-06139-t002:** Scoping review inclusion and exclusion criteria.

Criteria	Include	Exclude
Date	2005 to February 2022	Pre-2005
Language	English	Non-English language
Country	High-income ^1^ countries	Low-income countries, Whole regions (e.g., European Union), Global data
Publication, Study Design	Systematic review, Umbrella review, Aggregate report or evaluation	Thesis, Narrative review, Editorial, Discussion, Protocol, Conference abstract
Population	Human studies, Universal or population groups, including priority populations, any age or sex	Animal or In vitro studies
Concept 1: Prevention	Primary and/or secondary prevention, (e.g., Smoking, Nutrition, Alcohol, Physical activity, High cholesterol etc.)	Primordial, tertiary, or quaternary prevention, Pharmacotherapy for treatment of established disease, medical devices, COVID-19
Concept 2: Economic	Full economic evaluation (cost-effectiveness analysis, cost–benefit analysis, cost–utility analysis, cost–consequence analysis, cost-minimisation analysis), Return on investment, Value for money, Social return on investment	Methodological paper or framework, Partial economic evaluation (e.g., costing study)
Context	Public health service/setting/local public health services as the provider of Concept 1	National- or state-level strategies/initiatives (e.g., regulation, taxation, mass media campaigns, transport, infrastructure, urban planning), Privatised health systems, Workplaces

^1^ Maximum of 25% of studies in the review from middle-income countries.

**Table 3 ijerph-20-06139-t003:** Criteria for evaluating the economic evidence from the systematic reviews.

Assessment Categories	Criteria
Cost-effective	5 or more studies were included in the systematic review ≥70% of studies or interventions were cost-effective or median ICERs < USD 50,000 or GBP 30,000
Not cost-effective	5 or more studies were included in the systematic review ≥70% of studies or interventions were not cost-effective or median ICERs > USD 50,000 or GBP 30,000
Lack of evidence	<5 studies were included in the systematic review
Unclear	Findings from the studies were mixed or inconclusive or the studies included were too heterogeneous to draw conclusions

ICER, Incremental cost-effectiveness ratio.

**Table 4 ijerph-20-06139-t004:** Characteristics of included systematic reviews.

First Author, Year	No. of Countries Included	Date Range of Publications	Aim of the Systematic Review	Population Included	Sample Size of Included Studies, Median (Min, Max)	Prevention Type (Primary, Secondary)
**Mental Health**						
Le, 2021 [30]	20 ^a^	2007 to 2020	To evaluate the cost-effectiveness of mental health promotion and prevention interventions	Universal	407 ^b^ (51, 12,864)	Primary
Park, 2013 [31]	3	2000 to 2012	To evaluate the cost-effectiveness of physical health promotion interventions	Adults and older adults with clinically diagnosed mental health disorders	232 ^b^ (87, 2160)	Secondary
Soneson, 2020 [18]	1	2007 to 2017	To evaluate the cost-effectiveness of psychological interventions for psychotic experiences ^c^	Adolescents and adults with psychotic experiences	196 (196, 196)	Secondary
**Obesity**						
Lehnert, 2012 [32]	7 ^a^	2006 to 2017	To evaluate the long-term (≥40 years) cost-effectiveness of obesity prevention interventions	Universal	NR	Primary
**Type 2 diabetes**						
Glechner, 2018 [33]	8 ^a^	2003 to 2016	To evaluate the cost-effectiveness of lifestyle intervention for the prevention of T2D and secondary diseases ^c^	Adults with pre-diabetes	NR	Primary
Li, 2015 [34]	10 ^a^	1998 to 2014	To evaluate the cost-effectiveness of diet and physical activity promotion for the prevention of T2D	Adults and older adults at increased risk of T2D	3234 ^b^ (552, 3887)	Primary
Zhou, 2020 [35]	9 ^a^	2008 to 2017	To evaluate the cost-effectiveness of T2D prevention interventions	Adolescents, adults, and older adults at high-risk of T2D and universal	NR	Primary
**Dental caries**						
Anopa, 2020 [36]	6 ^a^	1986 to 2017	To review economic evaluations on primary caries prevention interventions	Pre-school children	964 ^b^ (161, 209,285)	Primary
Fraihat, 2019 [37]	8 ^a^	1976 to 2018	To evaluate the cost-effectiveness of primary caries prevention interventions for dental diseases ^c^	Pre-school and primary aged children	419 ^b^ (51, 209,285)	Primary
**Public health**						
Ashton, 2020 [22]	6	2007 to 2019	To evaluate SROI and SCBA evidence of public health interventions for health and well-being	Universal	NR	Primary
Banke-Thomas, 2015 [38]	11 ^a^	2005 to 2014	To assess studies where SROI has been applied in public health, lessons learnt, and recommendations for future	Universal	NR	Primary
Masters, 2017 [21]	6	1976 to 2015	To evaluate the return of investment of public health interventions	Universal	1454 ^b^ (123, 16,375)	Primary
Reeves, 2019 [39]	5	2000 to 2017	To review economic evaluations of strategies for enhancing the implementation of public health interventions and policies	Universal	NR	Primary
**Chronic disease**						
Dubas-Jakobczyk, 2017 [40]	11	2000 to 2015	To review the cost-effectiveness of health promotion and/or primary prevention programmes for chronic disease	Older adults	412 ^b^ (76, 33,152)	Primary
Gordon, 2007 [41]	7	1995 to 2005	To evaluate the cost-effectiveness of face-to-face health behaviour interventions for smoking, physical activity, diet, and alcohol for the prevention of chronic disease	Adults	NR	Primary
Mattli, 2020 [42]	5	2000 to 2018	To review the literature from RCT-based economic evaluations of physical activity interventions outside the workplace setting for chronic disease prevention	Adults and older adults	911 (51, 2140)	Primary
Pennington, 2013 [43]	2	2002 to 2006	To synthesise the evidence on cost-effectiveness of health-related lifestyle advice delivered by peer or lay advisors for chronic disease prevention ^c^	Adults	NR	Primary
Vos, 2011 [29]	1	2003	To evaluate the cost-effectiveness of preventive interventions for non-communicable diseases	Universal	NR	Primary, secondary
**Sexual health**						
Bloch, 2021 [44]	7	2000 to 2018	To synthesise the economic evidence on interventions for the prevention and management of sexually transmitted infections and HIV	Adolescents and adults	NR	Primary, secondary
**Immunisation**						
Boccalini, 2021 [45]	5	2013 to 2020	To evaluate the cost-effectiveness of influenza vaccination	Children	NR	Primary
**Smoking cessation**						
Cheung, 2017 [46]	1	2013 to 2016	To review the cost-effectiveness of eHealth smoking cessation interventions	Adults	NR	Secondary
Lee, 2019 [47]	5	1993 to 2016	To appraise the methodological quality and evaluate cost-effectiveness studies of inpatient smoking cessation programmes	Adults hospitalised with any conditions	433 ^b^ (224, 4404)	Secondary
Mahmoudi, 2012 [48]	8	2008 to 2010	To review the cost-effectiveness of non-nicotine therapies for smoking cessation, compare the types of models used, and determine if any variables impact on the cost-effectiveness	Adults	NR	Secondary
**Reducing alcohol**						
Kruse, 2020 [49]	1	2011	To evaluate cost-effectiveness of telemedicine for the management of alcohol abuse, addiction, and rehabilitation ^c^	Adults with alcohol use disorder	1,216,000 (1,216,000, 1,216,000)	Secondary
**Fractures**						
Ganda, 2013 [50]	4	2007 to 2011	To evaluate the cost-effectiveness of secondary preventions for osteoporotic fractures ^c^	Adults and older adults	1140 (349, 620,000)	Secondary
Wu, 2018 [51]	6	2007 to 2017	To evaluate the cost-effectiveness of fracture liaison services or secondary fracture preventive programmes	Adults and older adults	1000 ^b^ (100, 10,000)	Secondary

^a^ <25% from middle-income countries; ^b^ not all studies in review reported sample size; ^c^ clinical efficacy of the interventions was also evaluated. HIV, Human immunodeficiency virus; NR, Not reported; RCT, Randomised control trial; SCBA, Social cost–benefit analysis; SROI, Social return on investment; STI, Sexually transmitted infections; T2D, Type 2 diabetes.

**Table 5 ijerph-20-06139-t005:** Priority populations ^a^ of included systematic reviews.

First Author, Year	Indigenous ^b^	Culturally and Linguistically Diverse	Lesbian, Gay, Bisexual, Transgender, Queer or Questioning, Intersex, and/or Other Sexuality and Gender Diverse	Mental Illness	Low Socioeconomic Status	Disability	Rural, Regional, Remote
Anopa, 2020 [36]					X		
Ashton, 2020 [22]				X	X	X	
Banke-Thomas, 2015 [38]				X	X		
Bloch, 2021 [44]		X			X		
Boccalini, 2021 [45]							
Cheung, 2017 [46]							
Dubas-Jakobczyk, 2017 [40]						X	
Fraihat, 2019 [37]					X		
Ganda, 2013 [50]							
Glechner, 2018 [33]					X		
Gordon, 2007 [41]							
Kruse, 2020 [49]				X			
Le, 2021 [30]				X			
Lee, 2019 [47]				X		X	
Lehnert, 2012 [32]							
Li, 2015 [34]		X				X	
Mahmoudi, 2012 [48]						X	
Masters, 2017 [21]				X	X	X	
Mattli, 2020 [42]							
Park, 2013 [31]		X		X	X	X	
Pennington, 2013 [43]		X		X		X	
Reeves, 2019 [39]		X					X
Soneson, 2020 [18]		X		X			
Vos, 2010 [29]	X ^a^		X	X		X	X
Wu, 2018 [51]						X	
Zhou, 2020 [35]		X			X	X	

X indicates that the study sample included the specified priority population. ^a^ Priority populations identified in the National Preventive Health Strategy [4]; ^b^ Aboriginal and Torres Strait Islander people, Australia.

**Table 6 ijerph-20-06139-t006:** Economic evaluation methods, risk of bias assessment, and key findings.

First Author, Year	No. of Economic Evaluation Studies	Economic Analysis Method Used	Study Design	Risk of Bias Methodological Assessment Tool Used	Economic Outcomes	Key Economic Findings Reported by the Reviews
**Mental health**						
Le, 2021 [30]	65	CEA, CUA, ROI	30 RCTs, 29 simulation models, 2 quasi, 2 pre–post, 1 cross-sectional, 1 ecological	Quality of Health Economic Studies Instrument	QALYs DALYs, ICER	In children and adolescents (<18 years) (*n* = 23 studies): interventions targeted depression (*n* = 7), anxiety (*n* = 4), behaviour (*n* = 3), suicide (*n* = 4), eating disorders (*n* = 2), cannabis use (*n* = 1), maltreatment (*n* = 1), and general mental health (*n* = 1). In children and adolescents, screening plus psychological interventions at school and parenting interventions were the most cost-effective interventions. In adults (18 to 65 years) (*n* = 35 studies): interventions targeted depression (*n* = 11), suicide (*n* = 8), general mental health (*n* = 7), eating disorders (*n* = 2), psychosis (*n* = 2), substance use (*n* = 1), anxiety (*n* = 1), and panic disorder (*n* = 1). In adults, screening plus psychological interventions were shown to be cost-effective. In older adults (>65 years) (*n* = 7 studies): interventions targeted depression (*n* = 6), anxiety (*n* = 4), and general mental health (*n* = 1). The cost-effectiveness of mental health interventions in older adults is inconclusive due to limited evidence.
Park, 2013 [31]	11	CCA, CEA, CUA	8 RCTs, 2 simulation models, 1 pre–post	No tool used	Incremental cost per: successful quit, life year gained, QALY gained	Interventions targeted sedentary behaviour (*n* = 3), substance misuse (*n* = 3 studies), infectious diseases (*n* = 4), and smoking (*n* = 1). Physical activity interventions ranged from cost-effective for supervised walks (99.9% probability) and tailored exercise programmes (89.0%) to not cost-effective for facilitated support (57.0%). Substance abuse support programmes using case managers were not cost-effective. The cost-effectiveness of HIV interventions was gender specific or they were not cost-effective. The prevention or management of blood-borne disease using mobile specialist teams was evaluated, however, the cost-effectiveness of this intervention was unclear. Multistrategy smoking cessation programme in outpatient setting was cost-effective (74.0%).
Soneson, 2020 [18]	2	CEA, CUA	2 RCTs	Drummond Critical Appraisal of Economic Evaluations Checklist	Transition to psychosis averted, QALYs	CEA found routine care plus CBT had a 64% probability of being cost-effective at 18 months and 83% at 4 years compared with routine care. CUA found routine care plus CBT had an 83% probability of being cost-effective at 18 months and 86% at 4 years compared with routine care.
**Obesity**						
Lehnert, 2012 [32]	16	CUA	All simulation models	No tool used	QALY, DALY	Across the 16 publications, 21 behavioural and 12 community interventions were identified. For behavioural interventions, 16 interventions were cost-effective, 1 was cost-saving, and 5 were not cost-effective. For community interventions, 9 interventions were cost-effective and 3 were not cost-effective.
**Type 2 diabetes**						
Li, 2015 [34]	22	CEA	18 simulation models, 4 RCTs	Community Guide protocol for economic evaluations	CBR ICER per LYG, QALY saved, DALY averted	Fifteen of sixteen studies that reported cost per QALY saved indicated that combined diet and physical activity promotion interventions were cost-effective (median of USD 13,761). Three studies reported cost savings and two studies found the interventions to be cost-effective based on cost per DALY averted (AUD 21,195 and AUD 50,707 per DALY).
Glechner, 2018 [33]	14	CEA	8 simulation models, 6 RCTs	Drummond Critical Appraisal of Economic Evaluations Checklist	Costs per life year gained, costs per QALY, costs per DALY, costs per avoided diabetes-associated outcome	Across the 13 studies (14 articles), 11 studies found that lifestyle interventions are cost-effective compared with no interventions or usual care. Cost per QALY ranged from USD 1100–1300 over a lifelong time horizon and from USD 31,500–34,500 over a 3-year time horizon.
Zhou, 2020 [35]	28	CEA	20 simulation models, 8 RCTs	Guidelines for authors and peer reviewers of economic submissions to the British Medical Journal	ICER, cost saved	In high-risk individuals, lifestyle interventions were the most cost-effective interventions (median ICERs of USD 12,520 per QALY) followed by metformin interventions (USD 17,089 per QALY). Diabetes prevention programme was the most cost-effective type of lifestyle intervention compared with non-diabetes prevention programme (USD 6212 vs. USD 13,228).
**Dental caries**						
Anopa, 2020 [36]	16	CBA, CEA, CUA	7 simulation models 6 quasi, 2 RCTs, 1 cohort	CHEERS Checklist	ICER, ACER, B/C ratio, cost per carious surface averted, cost per incremental change in dmfs, cost per tooth saved, cost per child saved from caries experience, cost per child saved from extraction experience, number of avoided restorative or surgical treatment visits	Six of fifteen studies that conducted CEA found that a dental disease management programme, education programmes, fluoridated milk and milk–cereal, and five caries prevention interventions were cost-effective. Only 1 of 2 studies that conducted CBA demonstrated benefits of a combined hand hygiene and OH promotion programme. Only 1 study reported QALY as an outcome and found that home visits and telephone intervention were dominant and cost-saving compared with usual care.
Fraihat, 2019 [37]	19	CEA	10 RCTs, 9 simulation models	Drummond Critical Appraisal of Economic Evaluations Checklist	Decayed, missing, filled teeth, QALY, dental visits	Oral health promotion was found to be effective for reducing the costs in 97 of 100 interventions (95% CI 89–99%, I^2^: 99%, *p* = 0). Sub-group analyses by age group identified that oral health promotion interventions were effective in reducing incremental cost for children 6 years and older but were not cost-effective for children less than 6 years old.
**Public health**						
Ashton, 2020 [22]	40	SROI	39 case studies, 1 simulation model	Krlev et al.’s framework	Crude SROI ratio	Public health interventions were identified across the life course for the included studies which were stage 1: birth, neonatal period, post-natal period, and infancy (*n* = 2 studies); stage 2: childhood and adolescence (*n* = 17); stage 3: adulthood (main employment and reproductive years) (*n* = 8); and stage 4: older adulthood (*n* = 6), as well as studies across the life course (*n* = 7). Interventions during stage 1 targeted breastfeeding and crude SROI ranged from GBP 6.50 per GBP 1 invested to EUR 15.85 per EUR 1 invested. Interventions during stage 2 targeted general health and well-being, substance misuse, mental well-being, sexual health and teenage pregnancy, employment, physical activity, and anti-social behaviour. SROI ratios ranged from GBP 2 to GBP 9.20 per GBP 1 invested. Intervention during stage 3 targeted mental well-being, general health and well-being interventions, smoking, employment, and substance misuse. The SROI ratios ranged from GBP 0.66 to GBP 7 per GBP 1 invested. Interventions during stage 4 targeted mental well-being and isolation and loneliness. The SROI ratios ranged from GBP 1.20 to GBP 11 per GBP 1 invested. Across the life course interventions targeted general health and well-being, physical activity, and diet. SROI ratios ranged from GBP 44.56 per GBP 1 invested to GBP 2.56 per GBP 1 invested.
Banke-Thomas, 2015 [38]	40	SROI	39 case studies, 1 simulation model	Krlev et al.’s framework	SROI ratios	SROI evaluations were identified across a wide range of public health areas including health promotion (12 studies), mental health (11), sexual and reproductive health (6), child health (4), nutrition (3), healthcare management (2), health education, and environmental health (1 each). Across these studies there was a lack of agreement on who to include as beneficiaries and how to account for counterfactual and appropriate study-time horizons. Reported SROI ratios varied widely (1.1:1 to 65:1). Authors interpreted an SROI ratio > 1 as a worthwhile investment.
Masters, 2017 [21]	44	CEA, ROI	23 simulation models, 4 RCTs, 5 cohort matched control, 4 quasi, 2 mixed methods, 2 case studies, 1 cohort, 1 cross-sectional, 1 pre–post	NICE quality appraisal checklist for economic evaluations	CBR, ROI	Public health interventions were stratified by specialism including health protection interventions, health promotion interventions, and healthcare public health interventions. The median (range) ROI and CBR were 34.2 (−21.3 to 221) and 41.8 (1.2 to 167) for health protection interventions, 2.2 (0.7 to 6.2) and 14.4 (2.0 to 29.4) for health promotion interventions, while ROI was 5.1 (1.15 to 19.35) and no studies reported a CBR for healthcare public health interventions.
Reeves, 2019 [39]	14	CBA, CCA, CEA, CUA	12 RCTs, 2 simulation models	Drummond Critical Appraisal of Economic Evaluations Checklist, CHEERS Checklist	ICER, net monetary benefit statistics, CBR	Interventions targeted cancer, physical activity, combination of physical activity and diet, alcohol-related crime, and infectious diseases. Most studies (9 of 14) reported that public health interventions were cost-effective or had a positive cost–benefit ratio. Three studies reported that the interventions were not cost-effective while two studies made no conclusion regarding the cost-effectiveness.
**Chronic disease**						
Dubas-Jakobczyk, 2017 [40]	29	CBA, CCA, CEA, CUA	16 RCTs, 10 simulation models, 3 quasi	Drummond Critical Appraisal of Economic Evaluations Checklist	QALYs, the number of falls or number of falls prevented, avoidance of health service utilisation, and the number of femoral/hip fracture incidents prevented or time free of these fractures	Interventions targeted falls amongst the older population, disability, general health, physical activity, and oral health. Ten interventions which predominately (80%) focused on fall prevention were cost-effective or cost-saving. For 13 studies the cost-effectiveness of the intervention was unclear. Six studies concluded that the intervention was not cost-effective.
Gordon, 2007 [41]	64	CEA	31 RCTs, 23 simulation models, 3 quasi, 3 pre–post, 1 randomised trial, 1 cohort matched control, 1 cross-sectional study, 1 comparative study	Guidelines for authors and peer reviewers of economic submissions to the British Medical Journal	ICERs, per QALY gained, cost per LYS	Favourable cost-effectiveness was reported for smoking interventions (EUR 14,000 per QALY gained), physical activity interventions (EUR 53,119 per QALY gained), and multiple behaviour intervention in high-risk groups (cost-saving of EUR 40,094). The cost-effectiveness of alcohol and dietary interventions is unclear due to significant heterogeneity in the outcomes reported.
Mattli, 2020 [42]	12	CEA	12 RCTs	Consensus on Health Economics Criteria List	ICER per MET hour gained	Most interventions (18 of 22) were not cost-effective and reported an ICER above the authors’ cut-off benchmark of USD 0.44–0.63 per MET hour gained.
Pennington, 2013 [43]	3	CEA	All RCTs	Quality Assessment Tool for Quantitative Studies	QALYs, cost per additional mammogram, cost per LYS	Interventions targeted general chronic disease, T2D management, and breast cancer. A chronic disease self-management programme was found to be cost-effective (94% probability). Study findings indicated a telemedicine support programme for T2D was cost-effective (GBP 43,400/quality-adjusted life year). The cost-effectiveness of mammography promotion interventions varied depending on the target population.
Vos, 2011 [29]	150 interventions	CEA	All simulation models	ACE Priority Setting Checklist	DALYs	Specific topic areas that had ≥5 preventive interventions that both improved health and contributed to net cost savings or cost <USD 10,000 per DALY prevented (defined as “dominant” or “very cost-effective”) included: alcohol (7 of 9 interventions), mental disorders (7/11), tobacco (5/8), and other interventions (5/11). Specific topic areas that had ≥3 treatment interventions classified as dominant or very cost-effective included: mental disorders (5/10) and other treatment (3/6).
**Sexual Health**						
Bloch, 2021 [44]	31	CBA, CCA, CEA, CUA	30 simulation models, 1 pilot RCT	Guidelines for authors and peer reviewers of economic submissions to the British Medical Journal	MOAs, such as PID, ectopic pregnancy, or infertility, QALYs, monetary outcomes, or the number of patients cured	Studies analysed different screening options for chlamydia trachomatis, gonorrhoea, and HIV. Sixteen found chlamydia trachomatis screening is likely to be cost-effective for those <30 years of age. Nine studies concluded that chlamydia trachomatis screening was likely to be cost-effective under certain assumptions (e.g., appropriate uptake rate). However, the remaining 4 studies did not find STI screening to be cost-effective.
**Immunisation**						
Boccalini, 2021 [45]	8	CEA, CUA	All simulation models	No tool used	Cost/QALYs, cost/life year	All study authors concluded that childhood influenza vaccination with live attenuated vaccine, specifically the quadrivalent formulation, was cost-effective compared with the trivalent inactivated influenza vaccine or no vaccination (ICER: GBP 7234 vs. GBP 7989 per QALY gained).
**Smoking cessation**						
Cheung, 2017 [46]	2	CEA	2 RCTs	No tool used	Prolonged abstinence	Both studies reported the intervention to be highly cost-effective ranging from EUR 1500 for video-based counselling to EUR 5100 for an online programme and phone-based counselling to be paid for each additional abstinent participant compared with usual care.
Lee, 2019 [47]	9	CEA	4 RCTs, 5 simulation models	British Medical Journal’s checklist for reporting economic evaluations	The number of quitters, LYGs, QALYs, episode of non-fatal acute myocardial infarction, death, hospitalisation days	Smoking cessation programmes for hospitalised patients are highly cost-effective. No significant difference was found in the distribution of ICERs between studies that provide nicotine replacement therapy interventions compared with interventions without nicotine replacement therapy. ICERs for nicotine replacement therapy interventions ranged from dominant to USD 8354 per LY compared with dominant to USD 5568 per LY for interventions without nicotine replacement therapy.
Mahmoudi, 2012 [48]	10	CEA	10 simulation models	Drummond Critical Appraisal of Economic Evaluations Checklist	Relapse rate, ICER	Eight studies used a Markov BENESCO model for analysis, six of these studies found that varenicline dominated bupropion while the remaining two studies identified that varenicline was cost-effective. The 2 non-BENESCO model studies found varenicline to be cost effective with ICERs of USD 14,729 and USD 3303 per LYG.
**Reducing alcohol**						
Kruse, 2020 [49]	1	CEA	1 simulation model	No tool used	DALYs	Only 1 study investigated the cost-effectiveness of telemedicine for alcohol-related disorders. This study reported that the implementation of new eHealth technologies would improve the value of returns from EUR 1.08 per EUR 1 spent to EUR 1.62 in health-related value.
**Fractures**						
Ganda, 2013 [50]	5	CEA	2 simulation models, 1 quasi, 1 cohort study, 1 clinical audit	No tool used	Fracture prevented, fracture date, refracture rate, QALY gained	Four of four studies of interventions involving identification, assessment, and treatment of patients as part of the service were predicted or shown to be cost saving or cost-effective, with a cost of AUD 20,000 to AUD 30,000 per QALY gained. One study identifying and assessing people with a minimal trauma fracture, then making treatment recommendations to the primary care physician, without initiating treatment was found to be cost-effective, reporting cost per QALY gained was GBP 5740.
Wu, 2018 [51]	16	CBA, CEA	17 simulation models, 2 cohort matched controls	Drummond Critical Appraisal of Economic Evaluations Checklist	QALYs, DALYs, ICER	Overall, the FLS was shown to be cost-effective compared with usual care or no treatment, regardless of the programme intensity. The least expensive programmes such as mail-based interventions costing CAD 7 to CAD 8 per patient were associated with CAD 18,000 to CAD 22,000 in savings for a population of 1000 post-fracture patients. The upscaled implementation of FLS at 122 sites across the UK was estimated to prevent 31,000 fractures over the lifetimes of patients each year.

ACE, Assessing Cost-Effectiveness; BENESCO, Benefits of Smoking Cessation on Outcomes; CBT, Cognitive behavioural therapy; CCA, Cost–consequence analysis; CEA, Cost-effectiveness analysis; CER, Cost-effectiveness ratio; CHEERS, Consolidated Health Economic Evaluation Reporting Standards; CUA, Cost–utility analysis; DALY, Disability-adjusted life year; DMFS, Decayed, missing, filled surface; FLS, Fracture liaison service; ICER, Incremental cost-effectiveness ratio; MET, Metabolic equivalent of task; MOA, Major outcomes averted; National Institute for Health and Care Excellence, NICE; LYG, Life year gained; PID, Pelvic inflammatory disease; QALY, Quality-adjusted life year; RCT, Randomised control trial; ROI, Return on investment; SCBA, Social cost–benefit analysis; SROI, Social return on investment; T2D, Type 2 diabetes.

**Table 7 ijerph-20-06139-t007:** Categorisation of the cost-effectiveness of intervention by health area for the included systematic reviews (*n* = 26).

First Author, Publication Year of the Systematic Review (Number of Articles Included)	CE	Not CE	Lack of Evid.	Not Clear	Risk of Methodological Bias Assessment Reported in the Systematic Review
**Mental health**					
Le, 2021 [30] (*n* = 65 articles)				X	Most (92%) studies were assessed as fair to high methodological quality.
Park, 2013 [31] (*n* = 11 articles)				X	Not assessed.
Soneson, 2020 [18] (*n* = 2 articles)			X		All studies were of high methodological quality (met 87–90% of checklist components).
**Obesity**					
Lehnert, 2012 [32] (*n* = 16 articles, intervention targeted adults)	X				Not assessed.
Lehnert, 2012 [32] (*n* = 3 articles, intervention targeted children)			X		Not assessed.
**Type 2 diabetes**					
Glechner, 2018 [33] (*n* = 14 articles)	X				Most studies were of high methodological quality as only 2 checklist components were not met on average across the studies.
Li, 2015 [34] (*n* = 22 articles)	X				Assessed but results not reported.
Zhou, 2020 [35] (*n* = 28 articles)	X				Review only included studies with a quality score > 7 points (max. 13 points).
**Dental caries**					
Anopa, 2020 [36] (*n* = 16 articles)				X	Many (63%) of the studies had a quality appraisal score of ≥94%.
Fraihat, 2019 [37] (*n* = 19 articles)				X	Many (60%) studies had a quality appraisal score of ≥8 points (max. 10 points).
**Public health**					
Ashton, 2020 [22] (*n* = 40 articles)				X	Many (71%) studies received the highest quality appraisal rating.
Banke-Thomas, 2015 [38] (*n* = 40 articles)				X	Many (70%) studies received the highest quality appraisal rating.
Masters, 2017 [21] (*n* = 44 articles)				X	Many (71%) studies received the highest quality appraisal rating.
Reeves, 2019 [39] (*n* = 14 articles)				X	No single study met every reporting criterion and compliance was highly variable for the following quality measures: identification of the effects (29–79% of studies), measurement of effects (50–86%), and valuation of the effects (50–100%).
**Chronic disease**					
Dubas-Jakobczyk, 2017 [40] (*n* = 29 articles)				X	Most studies (86%) were assessed as “good” or moderate methodological quality.
Gordon, 2007 [41] (*n* = 64 articles)				X	Assessed but results not reported.
Mattli, 2020 [42] (*n* = 12 articles)		X			Most studies (83%) met ≥70% methodological quality checklist items.
Pennington, 2013 [43] (*n* = 3 articles)			X		Only publications assessed as “strong” in methodological quality were included in the review.
Vos, 2011 [29] (*n* = 150 articles)				X	Assessed but results not reported.
**Sexual health**					
Bloch, 2021 [44] (*n* = 31 articles)				X	Less than half (32%) of studies met ≥70% of quality checklist items.
**Immunisation**					
Boccalini, 2021 [45] (*n* = 8 articles)	X				Not assessed.
**Smoking cessation**					
Cheung, 2017 [46] (*n* = 2 articles)			X		Not assessed.
Lee, 2019 [47] (*n* = 9 articles)	X				More than half (56%) of studies met >70% of quality checklist items.
Mahmoudi, 2012 [48] (*n* = 10 articles)	X				Most (80%) studies met ≥90% of quality checklist items.
**Reducing alcohol**					
Kruse, 2020 [49] (*n* = 1 article)			X		Not assessed.
**Fractures**					
Ganda, 2013 [50] (*n* = 5 articles)	X				Not assessed.
Wu, 2018 [51] (*n* = 16 articles)	X				More than half (63%) of studies were assessed as high quality.

X indicates the assessment category that has been assigned to the review based on the strength of the evidence.

## Data Availability

Data are contained within the article.

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
