# Peer review of "A Scoping Review of Economic Evaluations to Inform the Reorientation of Preventive Health Services in Australia"

_ijerph, 2023, doi:10.3390/ijerph20126139_

Round 1
Reviewer 1 Report
1. In method and selection of articles, is there any reasoning or references about cut off 25% which used in this study?
2. In method, did authors use kappa to compared the review result betwen two reviewer in the screening stage?
3. please include the PRISMA-ScR checlist for this study.
only minor editing is required
Reviewer 2 Report
This study reviewed the cost-effectiveness of preventive health strategies to inform the reorientation of local public health services towards preventive health interventions that are financially sustainable.
The article was well-written, easy to follow, methodologically rigorous for a scoping review, and reports on an important topic. My only comment is that the PRISMA figure is blurry and needs to be corrected. Otherwise, this article is an excellent addition for this journal. I applaud the authors for their attention to detail.
Reviewer 3 Report
The aim of this study is to review the cost-effectiveness of preventive health strategies to inform the reorientation of local public health services towards preventive health interventions that are financially sustainable.
The authors did not clearly formulate the research problem. Please emphasize in the introduction why your research is important.
The article is very interesting and quite well prepared methodologically. Literature research was very well conducted. Moreover a peer-reviewed manuscript is clear, relevant to the field, and presented in a well-structured manner. Limitation, implication include necessary elements and information. The discussion refers to other research and has references to the theory that the author wants to develop. The figures and tables are appropriate.

Reviewer 4 Report
This paper aims to provide a review for cost-effectiveness of preventive health strategies. Authors search four databases to find articles that are published during the period between 2005 and 2022. JBI and PRISMA protocols are utilized, and 26 articles are included in the review. Findings imply that interventions related with obesity, type 2 diabetes, fractures, and smoking cessation are cost-effective.
I have read the paper with interest. Although the study reveals some insights and empirical evidence on the subject matter, I have concerns on the manuscript, which are listed below.
Comments:
· The manuscript does not provide justification for selection of 4 specific data bases for search. Why did the study not consider others such as Scopus, Web of Science, etc?
· Although it is argued that review focuses on more recent studies starting from the year 2005, there is no convincing discussion for the selection of this specific year. For instance, why did the study not cover last 20 years, i.e., 2002 to 2022?
· A methodological concern for the study is the choice of evaluation criteria. How did this study determine the threshold values used in Table 3?
· The study does not report country information for the reviewed articles. How many studies were conducted in Australia? Are these studies implying similar conclusions with other high income country studies?
· Are there any systematic differences in findings for middle income vs. high income countries?
· The study is motivated by a health policy strategy in Australia. However, the content does not provide any specific analysis for this country. The title of the paper may be revised accordingly, or the study should have specific section focusing on the case of Australia.
Round 2
Reviewer 4 Report
Revised version of the manuscript is acceptable for publication.